

# Iron-Bound Organic Carbon in Forest Soils: Quantification and Characterization

**Qian Zhao[1], Simon R Poulson[2], Daniel Obrist[3], Samira Sumaila[4, 5], James J. Dynes[4], Joyce M. McBeth[4, 5], Yu Yang[1]***

[1] {Department of Civil and Environmental Engineering, University of Nevada, Reno, Nevada, 89557}

[2] {Department of Geological Sciences and Engineering, University of Nevada, Reno, Nevada, 89557}

[3] {Division of Atmospheric Sciences, Desert Research Institute, Reno, Nevada, 89512}

[4] {Canadian Light Source, 44 Innovation Blvd, Saskatoon, SK, S7N 2V3, Canada}

[5] {Department of Geological Sciences, University of Saskatchewan, Saskatoon, SK, S7N 5E2, Canada}

* Correspondence to: Y. Yang (yuy@unr.edu)

## ABSTRACT

Iron oxide minerals play an important role in stabilizing organic carbon (OC) and regulating the biogeochemical cycles of OC on the earth surface. To predict the fate of OC, it is essential to completely understand the amount, spatial variability and characteristics of Fe-bound OC in natural soils. In this study, we investigated the concentrations and characteristics of Fe-bound OC in soils collected from 14 forests in the United States, and determined the impact of ecogeographical variables and soil physicochemical properties on the association of OC and Fe minerals. We found that Fe-bound OC contributed up to 57.8% of total OC (TOC) in forest soils. Atomic ratios of OC:Fe ranged from 0.56 to 17.7 with values of 1-10 for most samples, and these ratios indicate an importance of both sorptive and incorporative interactions. The fraction of Fe-bound OC in TOC ($f_{\text{Fe-OC}}$) was not related to the concentration of reactive Fe, which suggests that the importance of association with Fe in OC accumulation was not governed by the concentration of reactive Fe. Concentrations of Fe-bound OC and $f_{\text{Fe-OC}}$ increased with the latitude and reached peak values at a site with a mean annual temperature of 6.6 ˚C. Attenuated total reflectance-Fourier transform infrared spectroscopy (ATR-FTIR) and near-edge X-ray absorption fine structure (NEXAFS) analyses revealed that Fe-bound OC was less aliphatic than non-Fe-bound OC. Fe-



bound OC also was more enriched in $^{13}$C compared to the non-Fe-bound OC, but C/N ratios did
not differ substantially. In summary, $^{13}$C-enriched OC with less aliphatic carbon and more
carboxylic carbon was associated with Fe minerals in the soils, with values of $f_{\text{Fe-OC}}$ being
controlled by both sorptive and incorporative associations between Fe and OC. Overall, this study
demonstrates that Fe oxides play an important role in regulating the biogeochemical cycles of C
in forest soils, and uncovers the governing factors for the spatial variability and characteristics of
Fe-bound OC.

**1 Introduction**

Soil organic carbon (OC) in forests contributes 40% of the global carbon (C) mass, and is

a vital component of C biogeochemical cycles (Eswaran et. al., 1999). Global warming can
potentially accelerate the decomposition of forest soil OC, contributing to greenhouse gas
emissions (Steffen et al., 1998). Alternatively, forest soils can act as strong sinks for OC, if
appropriate management is implemented, such as forest harvesting and fire treatment (Eswaran et.
al., 1999; Johnson and Curtis, 2001). Understanding the fate and stability of forest OC is important
for evaluating and managing the global C cycle under the framework of climate change.

Currently, there is a large information gap concerning the stability and residence time of

OC, contributing to the problem that the residence time of OC (ranging from months to hundreds
of years) is a major source of uncertainty in modeling and prediction of C cycles (Schmidt et al.,
2011; Riley et al., 2014). Many concepts have been proposed to account for OC stabilization and
therefore residence times, including molecular recalcitrance, physical occlusion, and chemical
protection (Sollins et al., 1996; Krull et al., 2003; Baldock et al., 2004; Mayer et al., 2004;
Zimmerman et al., 2004; Schmidt et al., 2011). In general, the stability of OC is regulated by
biogeochemical reactions occurring at the interfaces between OC, minerals, and microorganisms,
and further knowledge about the mechanism for OC stabilization is critical for building up process-
based models to simulate and predict C cycles.

A number of lines of evidence suggest a key importance of iron oxide minerals in the

stabilization of OC (Kalbitz et al., 2005; Kaiser and Guggenberger, 2007; Wagai and Mayer, 2007).
Iron oxides have a relatively high sorption capacity for OC, with sorption coefficients for OC much
higher than that of other metal oxides (Kaiser and Guggenberger, 2007; Chorover and Amistadi,



2001). Wagai and Mayer (2007) reported Fe-bound OC concentrations in soils up to 22 mg g$^{-1}$ soil,
contributing up to 40% of total OC (TOC) for most forest soils. Similarly, Lalonde et al. (2012)
found that Fe-bound OC contributed 22% of TOC in sediments. Studies have shown that Fe
minerals protect OC from degradation and inhibit mineralization of OC (Baldock and Skjemstad,
2000; Kalbitz et al., 2005). There is, however, no systematic study on the occurrence of Fe-bound
OC across different forests and its governing factors.
The overall goals of this study were to investigate the spatial variability of Fe-bound OC
across forest soils and the controlling factors, and to study the characteristics of Fe-bound OC in
respect to physicochemical properties of soils. In this study, we first quantified the concentration
of Fe-bound OC across 14 forest soils in the United States and analyzed the spatial distribution
and influences of ecogeographical factors. Second, we investigated the impact of soil
physicochemical properties on the Fe-OC associations. Third, we studied molecular characteristics
of Fe-bound OC versus non-Fe-bound OC, including how Fe-OC association influenced the
chemical properties of OC and their stable isotope composition. Hence, this study provided a
systematic evaluation for the Fe-bound OC in United States forests, the influences of ecological
factors on the occurrence of Fe-bound OC, and the effects of association with Fe on the chemical
properties of OC.

**2. Methods & Materials**
**2.1 Chemicals and materials**
Reagents used for Fe reduction experiments include sodium bicarbonate (NaHCO$_3$: Sigma-
Aldrich, St. Louis, MO, USA), trisodium citrate dihydrate (Na$_3$C$_6$H$_5$O$_7$•2H$_2$O: Acros Organics,
New Jersey, USA), and sodium dithionite (Na$_2$S$_2$O$_4$: Alfa Aesar, Ward Hill, MA, USA). All
chemicals used were analytical grade.

**2.2 Soil sample collection, primary characterization and pretreatment**
Soil samples were collected from 14 forest sites in the United States (Obrist et al., 2011,
2012, 2015). The abbreviations and the basic information for the sites are summarized in Table 1.
More detailed information on the sites and sampling protocols can be found in previous
publications (Obrist et al., 2011, 2012, 2015). Soil texture was analyzed by an ASTM 152-type



hydrometer at the Soil Forage and Water Analysis Laboratory at Oklahoma State University
(Obrist et al., 2011). The soil pH was measured by mixing soil particle with deionized (DI) water
in a solid/solution ratio of 1:1 (Kalra, 1995). Soil samples used in the experiments in this study
were ground to < 500 μm and freeze-dried.

**Table 1**

**2.3 Total C (TC), TOC and stable C isotope analyses**
TC, TOC and stable C isotopic compositions of soil samples were analyzed using a
Eurovector elemental analyzer (Eurovector SPA, Milan, Italy) interfaced to a Micromass IsoPrime
stable isotope ratio mass spectrometer (Micromass UK Ltd., Manchester, UK). Acetanilide (71.09 %
C by weight) was used as a standard compound to establish a calibration curve between mass of C
and the m/z 44 response from the mass spectrometer. In this study, the concentration of TC and
TOC were expressed as weight %. Stable C isotope analyses were performed after the method of
Werner et al. (1999), with results reported in the usual delta notation in units of ‰ vs. Vienna Pee
Dee Belemnite (VPDB). For TOC analysis, soil samples were acidified with 1 M HCl with the
solution/solid ratio of 1 mL solution/0.5 g soil and heated at 100˚C for 1 hour. The treatment was
repeated three times until there was no further effervescence upon acid addition, after which the
samples were dried and analyzed.

**2.4 Nitrogen (N) analysis**
The N concentration of each sample was analyzed using a Eurovector elemental analyzer.
Acetanilide (10.36 % N by weight) was used as a standard compound to establish a calibration
curve between mass of N and the response of the thermal conductivity detector in the elemental
analyzer. Total N and non-Fe-bound N concentrations were measured before and after a Fe
reduction release treatment for each sample.

**2.5 Analysis of Fe-bound OC**
The concentration of Fe-bound OC was quantified by an established Fe reduction release
method, commonly known as DCB extraction involving sodium dithionite, citrate and bicarbonate
(Mehra and Jackson, 1960; Wagai and Mayer, 2007; Lalonde et al., 2012). The DCB extraction is





assumed to extract most free Fe oxides (i.e. goethite, hematite, ferrihydrite and others) existing in
soils, but should not extract structural Fe in clay minerals (Mehra and Jackson, 1960; Wagai and
Mayer et al., 2007; Lalonde et al., 2012). In this study, we followed the specific protocol detailed
in Lalonde et al. (2012). An aliquot (0.25 g) of soil was mixed with 15 mL of buffer solution at
pH 7 (containing 0.11 M bicarbonate and 0.27 M trisodium citrate), and then heated to 80°C in a
water bath. The reducing agent sodium dithionite was added to the samples with final
concentration of 0.1 M, and maintained at 80°C for 15 min. The samples were then centrifuged at
10,000 rpm for 10 min, the supernatant was removed, and the residual particles were rinsed using
5 mL of DI water. The rinse/centrifuge process was performed three times. The residual particles
were freeze-dried and analyzed for TC and TOC concentrations and $\delta^{13}C$ composition. The mass
of residual particles were used to calculate the OC concentration associated with non-Fe minerals.

The background release of OC during the heating process was measured following the

method in Lalonde et al. (2012), where sodium citrate and dithionite were replaced by sodium
chloride with the same ionic strength. An aliquot (0.25 g) of dry soil was mixed with 15 mL of 1.6
M NaCl and 0.11 M NaHCO₃, and heated to 80°C. Then 0.22 g of NaCl was added, and the solution
was maintained at 80°C for 15 min. The samples were then centrifuged at 10,000 rpm and rinsed
three times, and freeze-dried before analysis. The mass of residual particles was used to calculate
the concentration of OC released by heating to 80°C. In preliminary experiments, we found that
the solution pH increased rapidly during the heating-extraction process with bicarbonate and
sodium chloride only, and the increased pH values facilitated the release of additional OC. Hence,
we used a lower initial pH of 6 to compensate for the shift to higher pH during heating. To validate
the measurement for the concentration of OC released during heating, we also tested the release of
OC using a phosphate buffer (same ionic strength) in lieu of the bicarbonate buffer, which can
maintain a pH of 7 during heating. Our results showed that the concentration of OC released was
similar for both the bicarbonate and phosphate buffer extraction reactions (Supplementary Material,
Fig. S1).

**2.6 Quantification of reactive Fe**

The concentration of reactive Fe in soils was determined by analyzing the Fe released

during the DCB reduction process. After the reduction treatment, the supernatant of each sample
was filtered using a 0.2 μm syringe filter (cellulose acetate), and analyzed for Fe concentration by



inductively coupled plasma - atomic emission spectroscopy (Varian-Vista AX CCD, Palo Alto,
CA, USA) at an optical absorption wavelength of 259.9 nm.

**2.7 Attenuated total reflectance-Fourier transform infrared spectroscopy (ATR-FTIR)**
ATR-FTIR analysis to characterize the molecular composition of OC was performed for
original soil samples and residual soils after DCB extraction using a Thermo Scientific Nicolet
6700 FTIR (Waltham, MA). Dry soil samples were placed directly on the crystal and forced to
contact well with the crystal. Spectra were acquired at the resolution of 4 cm$^{-1}$ based on 100 scans.
Data collection and baseline correction were accomplished using OMNIC software version 8.3.103.

**2.8 Near-edge X-ray absorption fine structure (NEXAFS) analysis**
For further characterization of chemical structure of OM, carbon (1s) K-edge NEXAFS
analyses were performed for select soil samples, i.e. for soils with the highest and lowest values
of the fraction of Fe-bound OC to TOC. The soil particles were suspended in DI water and
deposited on an Au-coated silicon wafer attached to a Cu sample holder. Before analysis, samples
were dried in a vacuum desiccator. The X-ray-based experiments were performed on the Spherical
Grating Monochromator (SGM) beamline at the Canadian Light Source (Saskatoon, Canada)
(Regier et al., 2007). The energy scale was calibrated using citric acid (absorption at 288.6 eV).
Major technical parameters and set-up for the beamline include: X-ray energy ranges 250-2000
eV; 45 mm planer undulator; 1000 µm×100 µm spot size; silicon drift detectors (SDD); a titanium
filter before the sample; entrance and exit slit gaps of 249.9 µm and 25 µm (Gillespie et al., 2015).
Carbon 1s spectra were acquired by slew scans from 270 to 320 eV at 20 s dwell time and 20 scans
per sample on a new spot. For data normalization, $I_0$ was collected by measuring the scatter of the
incident beam from a freshly Au-coated Si wafer using SDD. Before the $I_0$ normalization, the pre-
edge baseline was adjusted to near zero to remove the scatter in the sample data (Gillespie et al.,

2015).

**3. Results and Discussion**
**3.1 Concentration of Fe-bound OC**
This study covered five major forest types in North America, including Spruce-Fir, Pine,
Oak, Chaparral, and Maple-beech-birch forests distributed between 29° and 47° N. For the 14



forest soils, TC concentrations ranged between 1.5±0.1 and 8.3±2.1% (all percentages given are
weight-based), and TOC concentrations ranged between 1.3±0.3 and 6.2±2.9%, which are
comparable to values previously reported for North American forests (Wagai and Mayer, 2007;
Wilson et al., 2013). Bicarbonate extraction-calibrated Fe-bound OC concentrations ranged from
0.3 to 1.9%, with the fraction of Fe-bound OC to TOC ($f_{Fe-OC}$) averaging 37.8±20.0% (Fig. 1,
Supplementary Material, Table S1). Forest HL (Maine) had the highest $f_{Fe-OC}$ of 57.8%, while
forests GS (Florida) and OR (Tennessee) had $f_{Fe-OC}$ values below detection limits (i.e., below 0.6%).
Based on an estimate that 1502 Pg (Pg=1×10$^{15}$ g) of OC is stored in terrestrial soils (Scharlemann,
et al., 2014), scaling up these results to a global estimate would yield 538.5±271.5 Pg of Fe-bound
OC residing in terrestrial soils. This is a very large pool, e.g. compared to an estimated 19-45 Pg
Fe-bound OC that reside in global surface marine sediments (Lalonde et al., 2012). As a major
component of the global C pool, Fe-bound OC in terrestrial soils therefore is expected to play a
key role in global C biogeochemical cycles.

**Fig. 1**

**3.2 Fe-OC association**
The values of $f_{Fe-OC}$ were influenced not only by the concentration of reactive Fe, but also
by the type of association between Fe and OC. In this study, the concentration of reactive Fe in
forest soils ranged from 0.1 mg g$^{-1}$ to 19.3 mg g$^{-1}$, which is relatively low compared to values of
reactive Fe of up to 180 mg g$^{-1}$ reported previously (Wagai and Mayer, 2007; Wagai et al., 2013)
(Fig. 2A). A Mollisol in forest sites MS (California) had the highest concentration of reactive Fe,
when a Spodosol in forest site GS (Florida) had the lowest reactive Fe concentration. To our
surprise, there was no significant correlation between $f_{Fe-OC}$ and the concentration of reactive Fe
(Pearson Correlation Coefficient $r$=–0.418, $p$=0.137, Fig. 2B). This suggests that the proportion of
Fe-bound OC is not strongly controlled by the reactive Fe concentration.
The OC:Fe molar ratio ranged from 0.56 to 17.7 for all 14 soils, with a value between 1
and 10 for 10 soils (Fig. 2A). Previous studies have suggested that the OC:Fe molar ratio can be
used as an indicator for the type of association between Fe oxides and OC, with lower values
indicating sorptive interactions, while higher values indicate incorporation of OC within Fe oxides
(Wagai et al., 2007; Guggenberger and Kaiser, 2003). The highest sorption capacity measured for





OC onto Fe oxide corresponds to an OC:Fe molar ratio = 1.0 (Kaiser and Guggenberger, 2006),
but by incorporation and co-precipitation of Fe oxide OC:Fe molar ratio can reach much higher
values (Guggenberger and Kaiser, 2003). With OC:Fe molar ratios generally between 1-10 for
most of the forest soils in this study, we propose that incorporation of OC into Fe oxides plays a
major role in the accumulation of Fe-bound OC exceeding sorption by at least a factor of 1 to 10
(Wagai and Mayer, 2007; Lalonde, 2012). However, for the HT (Michigan), HL (Maine) and TKF
(California) forest soils, the OC:Fe molar ratios were even higher than 10 with maximum value of
17.8 (Fig. 2A), implying that incorporation of OC into Fe oxides dominated at these sites. The
value of OC:Fe ratio was not related to the concentration of reactive Fe, and varied a lot for soils
with similar concentration of total reactive Fe (Fig. 2B). This indicates the type of interactions
between OC and Fe was not governed by the amount of Fe. The OC:Fe ratio is potentially regulated
by the mineral phases of Fe, as poorly-crystalline Fe oxide has a higher capacity to bind with OC
than crystalline Fe minerals (Eusterhues et al., 2014). When sorption dominates the interactions
between OC and Fe, OC:Fe can also be influenced greatly by the particle size and surface area of
Fe oxides (Gu et al., 1995). Further investigations are needed to determine the factors that control
the OC:Fe ratio, and also $f_{\text{Fe-OC}}$ values for soils. Nevertheless, the lack of (or poor) relationship
shown here between the concentration of Fe-bound OC and Fe concentrations demonstrates the
limitations associated with predicting and modeling the behavior of C in soils based on the Fe
concentrations in soils alone.

**Fig. 2.**

**3.3 Spatial variance and ecogeographical factors**
We analyzed the influences of ecogeographical factors on the occurrence of Fe-bound OC in forest
soils (Fig. 3). There was a significant correlation between the TOC concentration and latitude
(Pearson correlation coefficient $p$=0.619, $r$=0.018), a pattern commonly observed due to  lower
microbial activity and turnover rates of C at higher, colder latitudes (Davidson and Janssens, 2006).
Concentration of reactive Fe, if excluding soil MS in California, it was also significantly related
to latitude ($p$=0.824, $r$=0.001). Both concentrations of Fe-bound OC and $f_{\text{Fe-OC}}$ also were correlated
with latitude ($p$=0.523, $r$=0.053; $p$=0.525, $r$=0.054). Among our samples, soil in forest HL in



Maine, one of the three northern-most site with latitude of 45˚, had the highest $f_{Fe-OC}$ of 57.8%. In
forest GS in Florida with lowest latitude of 29.7˚, the $f_{Fe-OC}$ were below detection limits, possibly
due to the low concentration of reactive Fe (0.08 mg g$^{-1}$). Hence, increase in latitude both increased
concentrations of TOC in soil as well concentrations of Fe-bound OC, suggesting increased
interactions between Fe oxide and OC at higher latitudes. There were no clear trends in TOC or
Fe-OC interactions with longitude. For elevation, we separated two groups of samples, with one
group located below 1000 m (asl) and the other group above (mainly around 2000 and 4000 asl).
Concentrations of TOC and Fe-bound OC, however, were not significantly different between the
two groups. There were no clear trends with precipitation either, although others have reported
positive relationships between mean annual precipitation and soil TOC concentration at a global
scale (Amundson, 2001). Both concentrations of Fe-bound OC and $f_{Fe-OC}$ reached highest value
with mean annual temperatures at 6.6˚C. The increased annual mean temperature increased the
concentration of Fe-bound OC and $f_{Fe-OC}$ when it was below 6.6˚C, then decreased when annual
mean temperature was over 6.6˚C. Our results therefore suggest a possibly ideal temperature range
where Fe-OC interactions may be particularly pronounced. Finally, the study covered 7 major soil
orders, i.e. Alfisols (sample number n=3), Spodosols (n=4), Mollisols (n=1), Inceptisols (n=2),
Entisols (n=2), Gelisols (n=1), and Ultisols (n=1). Although there are limited replications in many
of these soil orders, highest concentration of Fe-bound OC were observed in Spodosols. Regarding
$f_{Fe-OC}$, the ratios also were highest in Spodosols, possibly indicating a particular importance of Fe-
bound OC in this soil type which occupies 3.5% of US land areas and 4% global ice-free land (Soil
Survey Staff, 1999). However, due to the limited number of samples for each soil order, these
findings warrant further confirmation.

**Fig. 3**

**3.4 Impact of soil physicochemical properties on Fe-OC association**

Soil texture can potentially influence the accumulation of Fe-bound OC. Figure 4

demonstrates that the fraction of non-calibrated Fe-bound OC showed a significant positive
correlation vs. fraction of sand ($r$=0.72, $p$<0.001), and negative correlations vs. fraction of silt ($r$=–
0.697, $p$<0.001) and fraction of clay ($r$=–0.616, $p$<0.001). There were similar correlations between





labile OC, and the fraction of sand ($r$=0.72, $p$<0.001), silt ($r$=0.72, $p$<0.001) and clay ($r$=0.72,
$p$<0.001). However, the calibrated Fe-bound OC had no significant correlation vs. any of the
texture fractions. These correlations indicate that the labile OC was mainly associated with the
sand component of forest soils, but that the soil texture did not affect the Fe-bound OC. There has
been debate on the relative roles of sand, clay and silt in the stabilization of OC in soil (Percival et
al., 2000; Six et al., 2002; Eusterhues et al., 2005; Vogel et al., 2014). Eusterhues et al. (2005)
found a relationship between the resistance of organic matter to oxidative degradation and the clay
concentration in soils, suggesting the importance of clay minerals in the stabilization and
accumulation of soil OC. In contrast, Percival et al. (2000) found that the clay mineral fraction
explained little of the variation in the accumulation of OC across a range of soil types in New
Zealand. Vogel et al. (2014) found that less than 20% of clay mineral surfaces were covered by
the sorption of OC, indicating that a limited proportion of clay mineral surface contributed towards
the stabilization of OC. Our results suggest that the Fe oxide-mediated stabilization of OC was not
related to the size/aggregation-based process, although the labile carbon concentrations increased
with the fraction of sand in the soils.

**Fig. 4**

The Fe-OC association can also be influenced by the soil pH, which affects the mineral
phases of Fe oxides and their surface charge, and their interactions with OC. For our soil samples,
the soil pH ranged from 4.1 to 6.3, similar to measurements by Wagai and Mayer (2007) for North
America soils. There was no significant correlation between the $f_{Fe-OC}$ and soil pH, e.g. the HL
(Maine) soil with pH of 4.4 had the highest $f_{Fe-OC}$ of 57.8%, while the TS(II) (Washington) soil
with a similar pH of 4.5 only had a $f_{Fe-OC}$ of 7.4%. For soils with pH ranging from 4.9 to 5.8, $f_{Fe-OC}$
did not change correspondingly. Contrastingly, values of OC:Fe molar ratios were significantly
influenced by the soil pH; expect for one outlier sample of TS(II) (Washington) soil, there was a
significant negative correlation between the OC:Fe molar ratio and soil pH ($r$=-0.477, $p$=0.09)
(Supplementary Material, Fig. S2). This may be due to the lower pH values favoring the
complexation and precipitation of Fe with OC, while higher pH favors sorptive interactions
between Fe minerals and OC (Tipping et al., 2002). If comparing samples with similar pH, the
soils with higher TOC had higher OC:Fe molar ratios, e.g. the GS soil (TOC = 1.1%) with pH of



4.7 had an OC:Fe molar ratio = 8.5, while the HT (Michigan) soil (TOC = 3.0%) with similar pH
of 4.7 had an OC:Fe molar ratio = 17.1. This was consistent with the concept model that the form
of Fe was dominated by the Fe-OC co-precipitated complex with relatively higher OC supply
(Schwertmann et al., 1986).

**3.5 Molecular characteristics of Fe-bound OC**
The chemical composition of Fe-bound OC can be substantially different from non-Fe-
bound OC (Adhikari and Yang, 2015) with broad implications on the C biogeochemical cycles,
although such differences so far have received limited attention. We analyzed the difference in
chemical composition of Fe-bound OC compared to non-Fe-bound OC using ATR-FTIR analysis
(Fig. 5). Overall, there were limited fingerprint peaks for OC, because of the low concentration of
TOC and technical challenge for analyzing whole soil particles with FTIR (Calderon et al., 2011;
Simonetti et al., 2012). Reeves (2012) demonstrated that FTIR analysis of mineral soils in the
ranges of 1600-1750 and 2800-3000 cm$^{-1}$ only can be used to study OC. Peaks in the range of 500-
1200 cm$^{-1}$ indicate the presence of clay or other Fe/Al minerals (Fig. 5) (Madejova, 2003; Harsh
et al., 2002; Parikh et al., 2014), such as kaolinite or montmorillonite at 850-1200 cm$^{-1}$ (Madejova,
2003). Absorption at 850-1200 cm$^{-1}$ can also be due to the presence of polysaccharides, but
definitive identification of polysaccharides is not possible in the presence of minerals (Senesi et
al., 2003; Tandy et al., 2010). The spectra in the range of 1600-1750 cm$^{-1}$ normally contain
fingerprint peaks for functional groups of amides, carboxylates and aromatics (Parikh et al., 2014),
but we did not detect any significant peaks in this range. In the range of 2800-3000 cm$^{-1}$, there
were no significant peaks for the original soil samples, but after Fe extraction we detected
significant peaks at 2850 and 2930 cm$^{-1}$, which are characteristic for the presence of aliphatic
carbon. The substantial differences in spectra before and after Fe extraction indicate that aliphatic
OC was enriched in the residual soils after extraction. Other functional groups, such as aromatic
carbon and hydrophilic functional groups, were more strongly associated with Fe minerals and
removed during the Fe extraction, as hydrophilic functional groups can form inner-sphere
coordination complexation with iron oxides, and aromatic carbon has electron donor-acceptor
interactions with iron oxides (Gu et al., 1995; Axe and Persson, 2001).

**Fig. 5**




335   Furthermore, we analyzed the C 1s NEXAFS spectra of two original, non-extracted soils

with the highest and lowest values of $f_{\text{Fe-OC}}$, i.e. HL (Maine) ($f_{\text{Fe-OC}}$=57.8%) and OR (Tennessee)
($f_{\text{Fe-OC}}$ non-detectable) (Supplementary Material, Fig. S3). Three major fingerprint peaks were
detected for both soils, including peaks at 285.3, 287.0 and 288.7 eV, which are corresponding to
aromatic carbon, aliphatic carbon and carboxylic carbon, respectively (Schumacher et al., 2005;
Solomon et al., 2005; Lehmann et al., 2008). The OR (Tennessee) soil had a more substantial
signal at 287.0 eV than the HL (Maine) soil, indicating a higher aliphatic carbon concentration in
the OR (Tennessee) soil compared to the HL (Maine) soil. Ratio of carboxylic carbon to aromatic
carbon (peak height) was 3.8 for HL (Maine) and 1.0 for OR (Tennessee), suggesting that the HL
(Maine) soil with higher $f_{\text{Fe-OC}}$ has relatively more carboxylic carbon compared to aromatic carbon.
Hence, the C1s NEXAFS spectra suggest that the soil with the higher $f_{\text{Fe-OC}}$ has higher
concentration of carboxylic C, while the soil with the lower $f_{\text{Fe-OC value}}$ has a higher aliphatic C
concentration. This result is consistent with the comparison of ATR-FTIR spectra in soils before
and after Fe extraction, providing evidence that Fe oxides are mainly associated with more
hydrophilic and carboxylic carbon, while non-Fe-bound OC was more aliphatic.

351   To further investigate the relationships between soil OC and Fe minerals, we analyzed the

stable C isotopic compositions ($\delta^{13}$C) of Fe-bound vs. non-Fe-bound OC (i.e., the residual OC
after DCB extraction). The $\delta^{13}$C for original soil samples ranged from –24.5‰ to –27.5‰, and the
values for non-Fe-bound OC were –25.1‰ to –28.0‰. The $\delta^{13}$C for Fe-bound OC was calculated
by combined isotope-mass balance (equation (1))
$\delta^{13}C_{\text{TOC}} \times \text{TOC} = \delta^{13}C_{\text{labile}} \times OC_{\text{labile}} + \delta^{13}C'_{\text{Fe-OC}} \times OC'_{\text{Fe}} + \delta^{13}C_{\text{non-Fe-OC}} \times OC_{\text{non-Fe}}$    (1)
where TOC is the concentration of total organic carbon, $OC_{\text{labile}}$ is the concentration of labile OC
(extractable by bicarbonate buffer), $OC_{\text{non-Fe}}$ is the concentration of non-Fe-bound OC (residual
OC after Fe extraction), and $OC'_{\text{Fe}}$ is the concentration of Fe-bound OC (excluded the labile OC);
$\delta^{13}C_{\text{TOC}}$ is $\delta^{13}$C for bulk OC, $\delta^{13}C_{\text{labile}}$ is $\delta^{13}$C for labile OC, $\delta^{13}C'_{\text{Fe-OC}}$ is $\delta^{13}$C for Fe-bound OC,
$\delta^{13}C_{\text{non-Fe-OC}}$ is $\delta^{13}$C for non-Fe-bound OC. However, it is difficult to directly resolve the $\delta^{13}C_{\text{labile}}$
and $\delta^{13}C'_{\text{Fe-OC}}$ using this equation. We simplified it to equation (2):
$\delta^{13}C_{\text{Fe-OC}} = \dfrac{(\delta^{13}C_{\text{TOC}} \times \text{TOC} - \delta^{13}C_{\text{non-Fe-OC}} \times OC_{\text{non-Fe}})}{OC_{\text{Fe}}}$    (2)





where $\delta^{13}C_{Fe\text{-}OC}$ is $\delta^{13}C$ for Fe-bound OC (including the labile OC), $\delta^{13}C_{TOC}$ is $\delta^{13}C$ for bulk OC,
$\delta^{13}C_{non\text{-}Fe\text{-}OC}$ is $\delta^{13}C$ for non-Fe-bound OC, TOC is the concentration of total organic carbon, $OC_{non\text{-}}$
$_{Fe}$ is the concentration of non-Fe-bound OC, and $OC_{Fe}$ is the concentration of Fe-bound OC. The
$\delta^{13}C$ for Fe-bound OC was heaviest for the TKF (California) soil with a value of –23.0‰, and the
lightest for the GS (Florida) forest at –27.0‰. Across all study sites, Fe-bound OC was relatively
enriched in $^{13}C$ (1.5±1.2‰ heavier) compared to the non-Fe-bound OC.  However, there is also a
contribution of labile OC to the Fe-bound OC, where labile OC is the OC extracted during the
dithionite-absent extraction described earlier). The $\delta^{13}C$ value for labile OC can be calculated using
equation (3):
$$\delta^{13}C_{labile}=\frac{(\delta^{13}C_{TOC}\times TOC-\delta^{13}C_{non\text{-}labile}\times OC_{non\text{-}labile})}{OC_{labile}}$$    (3)
where $\delta^{13}C_{labile}$ is $\delta^{13}C$ for labile OC, $\delta^{13}C_{TOC}$ is $\delta^{13}C$ for bulk OC, $\delta^{13}C_{non\text{-}labile}$ is $\delta^{13}C$ for non-labile
OC, $OC_{non\text{-}labile}$ is the concentration of non-labile OC, and $OC_{labile}$ is the concentration of labile OC.
Calculated values of $\delta^{13}C_{labile}$ range from -23.4% to -30.3%, and were lighter than the values for
$\delta^{13}C_{Fe\text{-}OC.}$ Although it is not reliable to quantitatively calculate the $\delta^{13}C$ for Fe-bound OC
subtracting the influences of labile OC, these results indicate that the true value for $\delta^{13}C_{Fe\text{-}OC}$ should
be even somewhat heavier than the results presented in Fig. 6.

Our results demonstrate that Fe-bound OC was enriched in $^{13}C$ compared to the non-Fe-

bound OC in forest soils, which is consistent with results for sediments, where Fe-bound OC was
1.7±2.8‰ heavier than non-Fe-bound OC (Lalonde et al., 2012) (Fig. 6A).  Wang et al. (1998)
have shown that $^{13}C$-enriched organic matter in sediments was enriched with O and N (due to the
presence of compounds such as proteins and carbohydrate groups), while the lipid fraction was
relatively $^{13}C$-depleted. Similarly, compound-specific isotopic analyses have shown that O/N-rich
constituents, such as cellulose, hemi-cellulose and amino acids, are $^{13}C$-enriched compared to
hydrocarbons (Glaser, 2005), and these $^{13}C$-enriched O/N-rich compounds can associate with Fe
oxide extensively through inner-sphere coordination interactions (Parikh et al., 2014). The value
of $\Delta^{13}_{FeOC\text{-}nonFeOC}$ (= $\delta^{13}C_{Fe\text{-}OC}$ - $\delta^{13}C_{non\text{-}Fe\text{-}OC}$) (difference in $\delta^{13}C$ for Fe-bound OC and non-Fe-
bound OC) was inversely correlated with the molar ratio of OC:Fe ($r$=-0.53, $p$=0.05, Fig. 6B).
These relationships suggest that the enrichment in $^{13}C$ was to some degree related to the OC:Fe
ratio, with lower OC:Fe leading to higher enrichment in $^{13}C$. As discussed previously, lower OC:Fe
ratios indicate an increased contribution from sorptive interactions of OC with Fe minerals as
compared to incorporation of OC within iron oxides and OC, and these sorptive interactions



between O/N-rich organic compounds and Fe oxide results in the enrichment of $^{13}C$ of Fe-bound OC vs. non-Fe-bound OC. Previous studies have attributed the stability of relatively labile and reactive compounds, such as amino acids and sugars, to their interactions with minerals (Schmidt et al., 2011), and our results demonstrated the importance of sorption to Fe minerals in increasing the stability of relatively reactive labile compounds.

**Fig. 6**

Nitrogen (N)-containing functional groups are potentially important for the association between OC and Fe oxides, although the concentrations of N are much lower than C. The bulk soil contained 0.05-0.45 % N, while the non-Fe-bound component (i.e. the residual solid after DCB extraction) contained 0.06-0.32 % N. Concentrations of Fe-bound N, calculated by difference, ranged up to 0.13 %. However, it is important to note that this number is based without a calibration for labile N that may be removed by the dithionite-free DCB extraction (data not available). There were significant correlations between C and N concentrations for both bulk soils ($r$=0.847, $p$<0.001: Supplementary Material, Fig. S4) and the non-Fe-bound residual components ($r$=0.858, $p$<0.001: Supplementary Material, Fig. S4), with molar C/N ratios of 14.2±2.6 and 13.7±2.3 for bulk and non-Fe-bound OC, respectively. These C/N values are essentially identical to a previously observed molar C/N ratio = 14.3 for a large set of world-wide soils samples (Cleveland et al., 2007), and a molar C/N ratio = 14.4 for OC-rich samples in China (Tian et al., 2010). This result suggests that C/N ratios for Fe-bound OC did not differ from that of non-Fe-bound OC, assuming that the labile carbon did not have a substantially different C/N ratio. Therefore, in contrast to the $^{13}C$ enrichment observed for Fe-bound OC, the interactions with Fe minerals did not affect the C/N ratio substantially.

## 4. Conclusion

Fe oxides are recognized as an important mineral phase regulating the amount and characteristics of OM in forest soils. The spatial variability of Fe-bound OC is governed by the geographical factors, such as latitude and annual mean temperature, and also the soil physicochemical properties.





Chemical composition of Fe-bound OC is substantially different from the rest compartment of soil
OC. Overall, this study provided a comprehensive investigation into the spatial variance of Fe-
bound OC in forest soils, its governing factors, and how the Fe-OC associations affect the chemical
characteristics of OC. As a substantial amount of OC was bound to Fe oxides, and association with
Fe affected the quality of OC, Fe oxides can regulate the biogeochemical cycles of carbon and its
response to climate change.

**Acknowledgements**
This research was supported by DOE grant DE-SC0014275 and University of Nevada-Reno Start-
up Fund. NEXAF research described in this paper was performed at the Canadian Light Source,
which is supported by CFI, NSERC, the University of Saskatchewan, the Government of
Saskatchewan, WED Canada, NRC Canada, and CIHR. Sample collection was supported by a
former EPA Science-To-Achieve-Results (STAR) grant R833378. We also acknowledge the
helpful comments from the editor and reviewers during the stage of quick reports.

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



**Figure Captions**

**Figure 1.** Concentrations of total carbon (TC), total organic carbon (TOC) and Fe-bound OC in 14 forest soils across the United States. Duplicate measurements were conducted for each of two plots in every forest site. Error bars represent standard deviation of measurements of four replicates for each forest site.

**Figure 2. A** Concentration of reactive Fe and OC:Fe molar ratio in US forest soils. **B** Relationship between the fraction of Fe-bound OC in TOC ($f_{Fe-OC}$)/ OC:Fe molar ratio and reactive Fe concentration in US forest soils.

**Figure 3.** Correlation between the TOC, reactive Fe, concentration of Fe-bound OC, $f_{Fe-TOC}$, OC:Fe and ecogeographical parameters including latitude, longitude, elevation (asl), precipitation (mean annual) and temperature (annual mean).

**Figure 4.** Correlation of the fractions of iron-bound organic carbon (uncalibrated and calibrated for loss of labile OC) and labile organic carbon vs. fractions of sand, silt, and clay in forest soils.

**Figure 5**. Attenuated total reflectance-Fourier transform infrared spectroscopy (ATR-FTIR) analysis for representative forest soils before (black line) and after Fe extraction (red line). All the spectra are background-calibrated. Among the 14 forest soils sampled in this study, we used five different forest soils, with $f_{Fe-OC}$ ranging 5.6-57.8%.

**Figure 6. A**. $\delta^{13}C$ of total organic carbon and non-iron bound organic carbon for 14 U.S. forest sites. **B**. Correlation between $\Delta^{13}_{FeOC-nonFeOC}$ and molar ratio of OC:Fe.



**Table 1 Information for the 14 forest sites studied (Obrist et al., 2011, 2012, 2015)**

| Forest ID | Abbr. | Location | Soil Order | Climate Zone | Precip.[a] (mm y⁻¹) | Temp[b] (°C) | LAT(°)[c] | LONG(°)[d] | Elevation (m asl) |
|---|---|---|---|---|---|---|---|---|---|
| Ashland | **AL** | Ashland, Missouri | Alfisols | Humid Continental | 1023 | 13.9 | 38.73 | -92.20 | 210 |
| Bartlett | **BL** | Bartlett, New Hampshire | Spodosols | Humid Continental | 1300 | 4.5 | 44.0 | -71.29 | 272 |
| Marysville | **MS** | Marysville, California | Mollisols | Mediterranean climate | 775 | 16.9 | 39.25 | -121.28 | 386 |
| Gainesville | **GS** | Gainesville, Florida | Spodosols | Humid Subtropical | 1228 | 21.7 | 29.74 | -82.22 | 50 |
| Oak Ridge | **OR** | Oak Ridge, Tennessee | Ultisols | Humid Subtropical | 1350 | 14.5 | 35.97 | -84,28 | |
| Little Valley (post-fire) | **LVF** | Little Valley, Nevada | Entisols | Highland Climate | 551 | 5.0 | 39.12 | -119.93 | 2010 |
| Little Valley | **LV** | Little Valley, Nevada | Entisols | Highland Climate | 550 | 5.0 | 39.12 | -119.93 | 2011 |
| Truckee (post-fire) | **TKF** | Truckee, California | Alfisols | Highland Climate | 569 | 6.0 | 39.37 | -120.1 | 1768 |
| Truckee | **TK** | Truckee, California | Alfisols | Highland Climate | 568 | 5.9 | 39.37 | -120.1 | 1767 |
| Niwot Ridge | **NR** | Niwot Ridge, Colorado | Alfisols | Highland Climate | 800 | 1.3 | 40.03 | -105.55 | 3050 |
| Hart | **HT** | Hart, Michigan | Spodosols | Humid Continental | 812 | 7.6 | 43.67 | -86.15 | 210 |
| Howland | **HL** | Howland, Maine | Spodosols | Humid Continental | 1040 | 6.7 | 45.20 | -68.74 | 60 |
| Thompson I | **TSI** | Ravensdale, Washington | Inceptisols | Highland Climate | 1141 | 9.8 | 47.38 | -121.93 | 221 |
| Thompson II | **TSII** | Ravensdale, Washington | Inceptisols | Highland Climate | 1140 | 9.8 | 47.38 | -121.93 | 220 |

a: annual precipitation; b: annual mean temperature; c latitude; d: longitude.









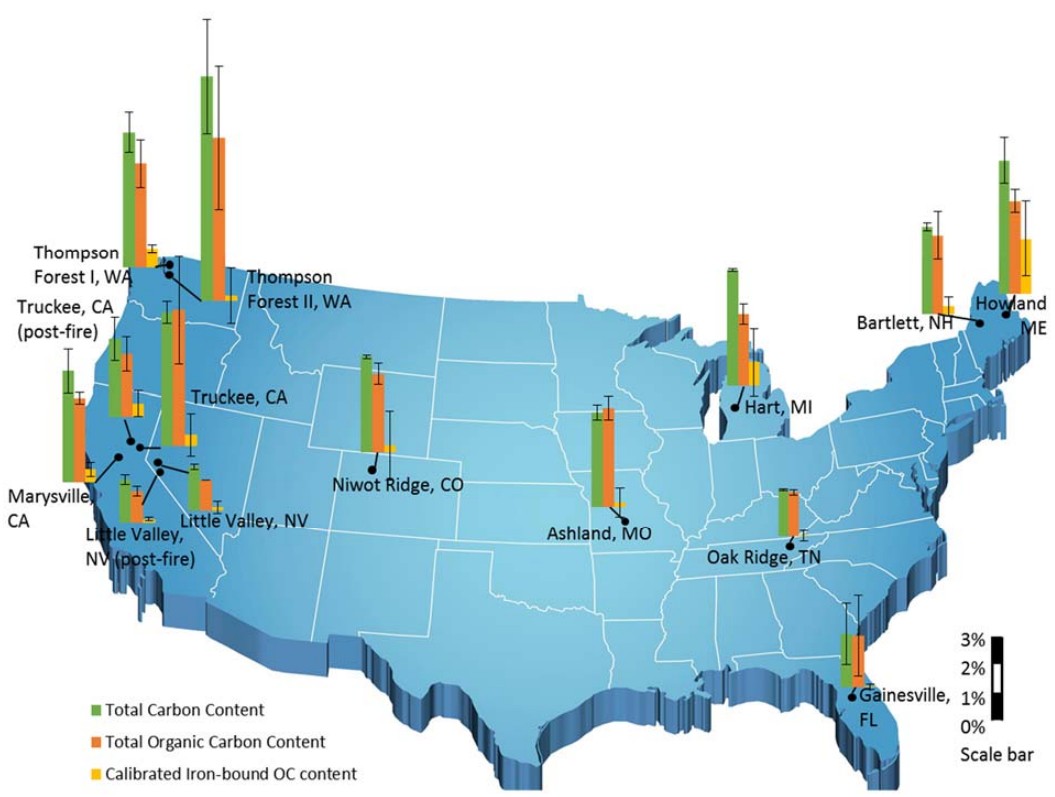


**Fig. 1**














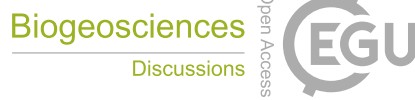



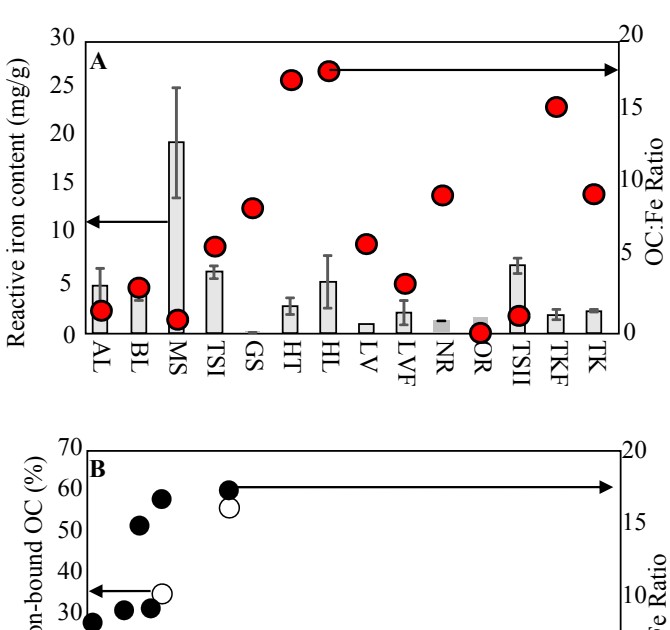


**Fig. 2**



















**Fig. 3**










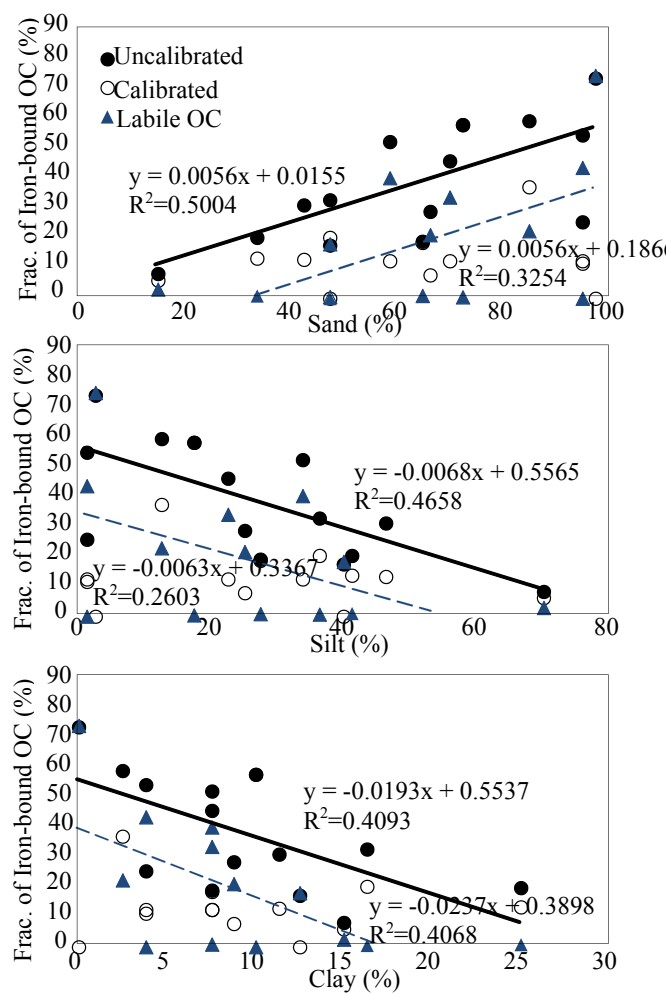


**Fig. 4**













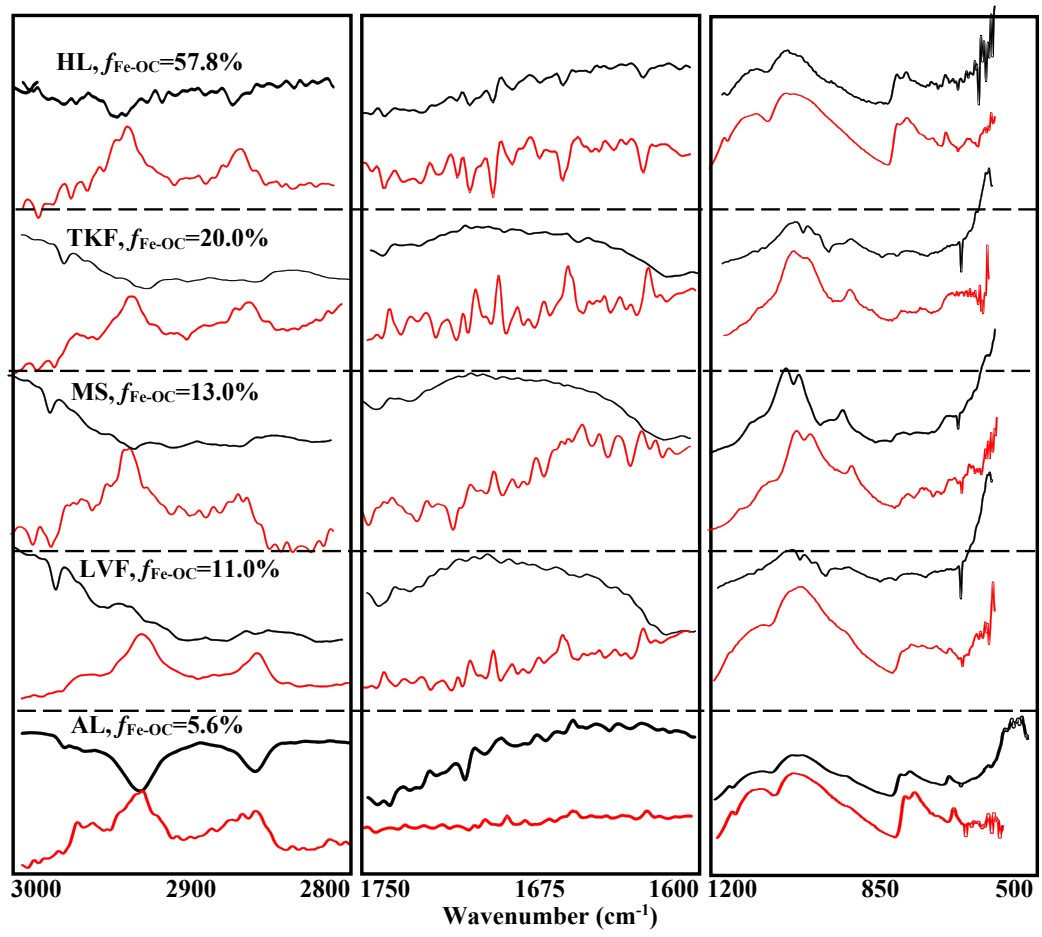


**Fig. 5**














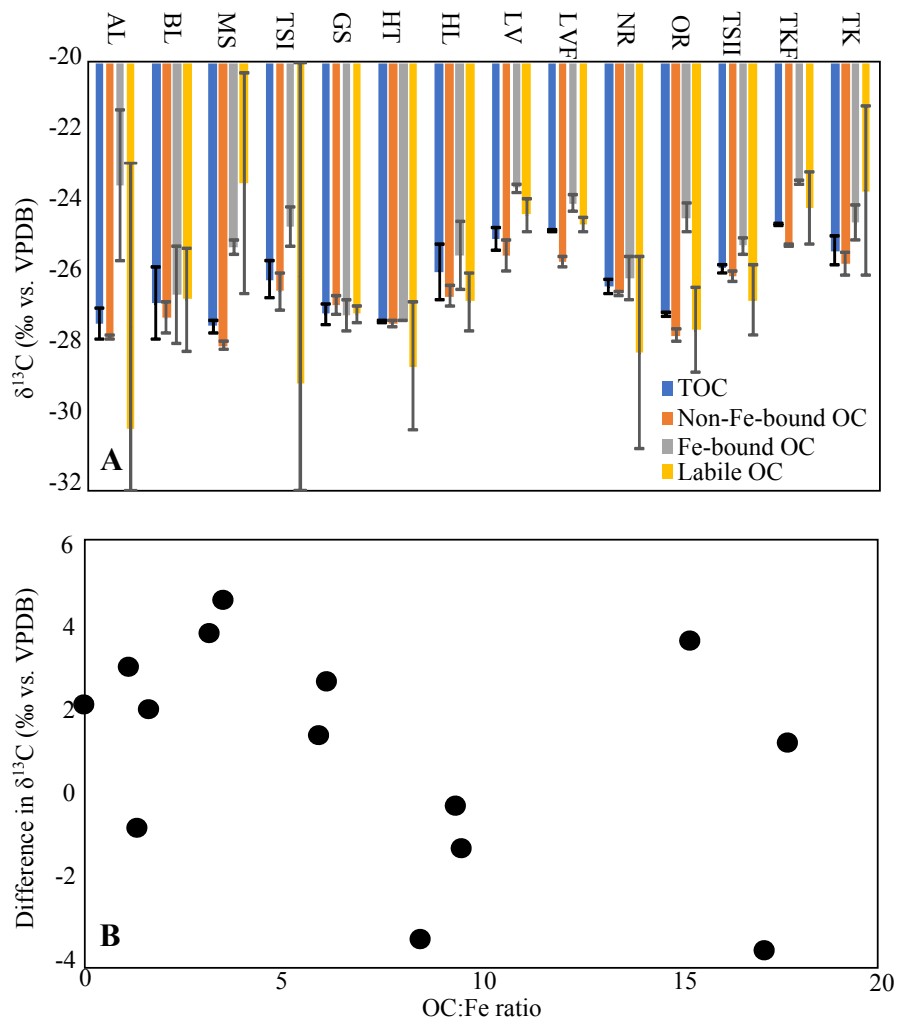


**Fig. 6**






