# Peer review of "Iron-Bound Organic Carbon in Forest Soils: Quantification and Characterization"

_Biogeosciences, 2015_

## Referee Comment (RC1) · Anonymous Referee #2 · 19 Feb 2016

The authors draw no conclusion from their data. This section needs to be revised. The current section limits itself to general statements and relatively vague summarizing sentences. In lines 309-311 the authors state that the chemical composition of Fe-bound can be substantially different and that this will have broad implications on the C biogeochemical cycles. They then demonstrate that there actually is a difference in chemical composition. The logical consequence hence is to now conclude which implications on the C cycle are to be expected from their findings. "Fe oxides can regulate the biogeochemical cycles of carbon and its response to climate change" is a very general statement and not a novel conclusion. In which way do they influence the C cycle considering that Fe-bound C is more aliphatic? "The spatial variability of Fe-bound OC is governed by the geographical factors, such as latitude and annual mean temperature, and also the soil physicochemical properties." is too general and

imprecise. In which way do they influence the spatial variability? How does this reflect on the C cycle?

---

## Short Comment (SC1) · 30 May 2016

Zhao et al investigated the role of iron oxides for organic carbon storage and preservation in north american forest soils. The study is based on chemical extractions and subsequent analysis of total iron and carbon. The different carbon fractions have than further been analysed for carbon isotopic as well as molecular composition. The study covers a number of different soils from various climatic regions, which is later considered in the discussion. Overall this is a nice study and the results provide good evidence for the emerging idea that iron minerals are one of many factors that govern organic matter preservation in soils. The study is well written, clearly organized and for the most part technically sound. The results are interesting to readers from soils sciences, biogeochemistry and partly for climate scientists working on soil-climate feedbacks. However, there a few minor and some major points that I like to see revised

before the paper can eventually be published.

Major remarks: 1) I did not check the references of Obrist et al to see how soils were sampled. However, since this is basic information for the present study I suggest that the sampling should briefly be described in the paper. For example, please explain if the different soil horizons were mixed to a composite sample. If not, which horizon was sampled and used in this study? In a spodosol, iron and organic matter are enriched in the deeper horizons, while being depleted in shallow ones. The chemical composition of both likely will be different in the different horizons, which will strongly affect the outcome of this study. 2) Unfortunately, the analytical section totally misses information on data quality. For the determinations of iron, nitrogen and carbon please add information on accuracy and precision. 3) I wonder if the chemical extractions influenced the molecular properties of the organic matter? Has this been tested by the authors? I suppose that it is possible for dithionite to change the properties oxidized functional groups. What about sulfurization of organic matter? Please provide a small discussion based on literature on the effect of reducing agents on organic matter and how that might possibly affect the results of the present study.

Minor remarks: Line 19. Having this number in the abstract somewhat skews the results of this study towards a higher importance of iron minerals than it really is. Fig. 1 gives a totally different picture. Maybe report median or range here.

Lines 90-91: Coarse material was not removed? What about roots?

Lines 119-121: Are manganese oxides also affected? Has Mn been measured in addition to Fe?

Lines 130-131: This is basically the ion exchangeable organic carbon. I probably missed it, but why was this done? Was the exchangeable organic carbon subtracted from the chemically extracted carbon? If not, please discuss why.

Line 151: Why was iron only determined at a single wavelength of 259.9 nm? Using

multiple wavelengths has the advantage of identifying interferences that might occur on one wavelength, which can than easily be corrected.

Lines 160-175 and 309-331: I am not an expert on NEXAFS and FTIR. I suggest the authors and the editor to refer to another reviewer with more expertise on these methods.

Line 180: Are numbers reported as one or two standard deviations of replicates? If so, how many (eg, n = 3)? Are this analytical or sampling replicates?

Lines 187-192: This part is speculative. Since not all soil types are covered the comparison might not hold up at the global level. For example, organic rich soils such as Histosols and Chernozems were not as much considered as they should be based on their global distribution. Further, since the North American distribution of soils differs from the global distribution in that Cambisols, Regosols and iron-rich chromic luvisols are underrepresented, this section should be removed.

Lines 202-205: Why was that a surprise? Was it one the main hypothesis of the study? If so, this might be stated in the introduction or methods. After all, shouldn't non-reactive iron store more organic carbon than reactive iron?

Line 214: How do you define "most of the"? It might be better to state "2/3 of the" or some equally (quantitative) phrase.

Lines 241-262: Climate actually controls the type of soil and the type of iron minerals directly through controlling the weathering of iron-bearing primary minerals and the formation of secondary iron-phases. So 6.6°C is probably the ideal condition for soil organic matter production and the formation OM-binding FeOOH minerals. I like to hear a statement by the authors on why "Fe-OC interactions may be particularly pronounced" at 6.6 °C. Can there really be a temperature control on this type of interaction? If so, please provide references for this.

Lines 252-254: This sentence is somewhat unclear. How does temperature increase

the amount of iron bound organic matter (see comment above)? Please re-write.

Line 258: I did expect this as these are relatively "old" soils. The long aging process accumulates old and possibly recalcitrant organic matter along with very stable iron minerals. . ..Also, aging results in more stable iron-organic matter associations as shown by Kaiser et al 2007 (doi:10.2136/sssaj2006.0189).

Lines 282-284: . . . and maybe the reduced chemical potential in small pores of clay-rich soils also limits microbial degradation of soils organic matter (Riedel & Weber 2016, doi:10.2136/sssaj2015.02.0085).

Lines 304-306: This short sentence is unclear. Please explain this 'concept model' in more detail.

Lines 347-349: A nice result. We have observed a similar effect using a complementary method (ultra-high resolution mass spectrometry) in a recent study (doi: 10.1016/j.orggeochem.2014.02.003). During a leaching test a soil column had turned anoxic and the reductive dissolution of iron- and manganese oxides lead to an increase in the abundance of aromatic compounds leaching from the soil columns. Besides polyphenolic compounds we also observed a release of iron-bound condensed aromatics leading me to belief that iron minerals probably play an important role in sequestering 'black carbon'-type compounds in soils. Do the NEXAFS or FTIR data of this study allow identification of this type of organic matter?

Lines 385 and 387. "O/N-rich". This is written like a ratio, but that is not meant here, right?

Lines 382-385. This probably indicates that the lipids are derived from microbial biomass which usually has a lighter isotopic value.

Lines 391-392: But it could also be vice versa. Suppose compounds with heavier isotopic composition preferentially sorb to iron.

Lines 392: Where was this "discussed previously"?

Lines 403-404. A reference would be good here.

Table 1: I had a hard time understanding table 1, because I am not familiar with the American soil taxonomy. In order to make the paper more readable to a readership beyond North America it might be a good idea if the authors could change to the international FAO soil classification standard, if this is possible. After all the paper has been submitted to a journal run by the 'European' Geophysical Union...

Figure 3: What was the hypothesis behind the linear regression analysis of TOC vs latitude? Did the authors suspect a linear relationship? If so, please explain why.

Thomas Riedel

---

## Editor Comment (EC1) · Dr Bol (Editor) · 7 Jun 2016

I confirm that the comment posted by Thomas Riedel is that of a nominated referee, hence two reviews have been received and on that basis the discussion can be closed

---

## Author Comment (AC1) · 24 Jun 2016

June 23, 2016

**Responses to Comments**

**Reviewers' comments are in black font and our responses are in *italicized* font.**

**Associate Editor**

I confirm that the comment posted by Thomas Riedel is that of a nominated referee, hence two reviews have been received and on that basis the discussion can be closed.

*Thank you for supporting our manuscript to move forward and all the helpful review. We have reviewed and responded to all the comments carefully.*

**Reviewer #1 (**Anonymous Referee #2)**

The authors draw no conclusion from their data. This section needs to be revised. The current section limits itself to general statements and relatively vague summarizing sentences. In lines 309-311 the authors state that the chemical composition of Fe-bound can be substantially different and that this will have broad implications on the C biogeochemical cycles. They then demonstrate that there actually is a difference in chemical composition. The logical consequence hence is to now conclude which implications on the C cycle are to be expected from their findings. "Fe oxides can regulate the biogeochemical cycles of carbon and its response to climate change" is a very general statement and not a novel conclusion. In which way do they influence the C cycle considering that Fe-bound C is more aliphatic? "The spatial variability of Fe-bound OC is governed by the geographical factors, such as latitude and annual mean temperature, and also the soil physicochemical properties." is too general and imprecise. In which way do they influence the spatial variability? How does this reflect on the C cycle?

*We will revise the conclusion to be more specific, according to the comments and suggestions:*

*Overall, this study provided a comprehensive investigation into the amount and characteristics of Fe-bound OC in forest soils as well as the impact of soil physicochemical properties. On average, Fe-bound OC contributed to 37.8% of TOC in forest soils, composing an important component of C cycles in terrestrial ecosystem. OC:Fe molar ratio ranged from 0.56 to 17.7, indicating the importance of both sorptive and incorporative interactions between Fe and OC. $f_{Fe-OC}$ increased with latitude, and reached the peak value for soils with annual mean temperature of 6.6°C, as a result of temperature dependence of Fe mineral phase and OC transformation. Combined studies of FTIR, NEXAFS, and $^{13}C$ analysis revealed that Fe-bound OC was less aliphatic, more carboxylic, and more enriched in $^{13}C$, compared to non-Fe-bound OC. Assuming Fe-bound OC is relatively stable, Fe oxides will serve as a long term storage of more hydrophilic and carboxylic OC, which would be otherwise relatively more available for microbial degradation.*

**Reviewer #2 (**t. riedel thomas.riedel@tu-braunschweig.de**)**

Zhao et al investigated the role of iron oxides for organic carbon storage and preservation in north american forest soils. The study is based on chemical extractions and subsequent analysis of total iron and carbon. The different carbon fractions have than further been analysed for carbon isotopic as well as molecular composition. The study covers a number of different soils from various climatic regions, which is later considered in the discussion. Overall this is a nice study and the results provide good evidence for the emerging idea that iron minerals are one of many factors that govern organic matter preservation in soils. The study is well written, clearly organized and for the most part technically sound. The results are interesting to readers from soils sciences, biogeochemistry and partly for climate scientists working on soil-climate feedbacks. However, there a few minor and some major points that I like to see revised before the paper can eventually be published.

*Thank you for the positive comments about our manuscript.*

Major remarks: 1) I did not check the references of Obrist et al to see how soils were sampled. However, since this is basic information for the present study I suggest that the sampling should briefly be described in the paper. For example, please explain if the different soil horizons were mixed to a composite sample. If not, which horizon was sampled and used in this study? In a spodosol, iron and organic matter are enriched in the deeper horizons, while being depleted in shallow ones. The chemical composition of both likely will be different in the different horizons, which will strongly affect the outcome of this study.

*This is a good suggestion. All the soils used in this study are from top layer (0-20 cm).*

*We will add more information about soil sampling in section 2.2:*

*More detailed information about the sites and sampling protocols can be found in previous publications (Obrist et al., 2011, 2012, 2015). Briefly, two replicate plots at each forest site were sampled. Top soils (0-20 cm) from all sites were collected using clean latex gloves and stainless steel sampling equipment. All samples were immediately transferred to plastic freezer bags and kept on ice before transportation to the laboratory.*

2) Unfortunately, the analytical section totally misses information on data quality. For the determinations of iron, nitrogen and carbon please add information on accuracy and precision.

*We will add the information about accuracy and precision information for analysis toward the end of session 2.3, 2.4 and 2.6:*

*All the analysis is based on standard curves with $R^2>0.99$. Detection limit for N is 0.2 mg/g soil. Average coefficient of variation for the analysis of N is 20.5%.*
*All the analysis is based on standard curves with $R^2>0.99$. Detection limit for C is 0.2 mg/g soil. Average coefficient of variation for the analysis of C is 20.2%.*
*All the analysis is based on standard curves with $R^2>0.99$. Detection limit for Fe is 0.04 mg/g soil. Average coefficient of variation for the analysis of Fe is 25.8%.*

3) I wonder if the chemical extractions influenced the molecular properties of the organic matter? Has this been tested by the authors? I suppose that it is possible for dithionite to change the properties oxidized functional groups. What about sulfurization of organic matter? Please provide a small discussion based on literature on the effect of reducing agents on organic matter and how that might possibly affect the results of the present study.

*Good point. Previous studies did not note the transformation of organic matter, upon the reaction with dithionite (Lalonde et al., 2012; Wagai and Mayer, 2007). The most possible reaction between dithionite and organic matter is the reduction of oxidized organic functional groups. Our recent study showed that dithionite could reduce quinone groups in natural organic matter (Adhikari et al., 2016). For our discussion, quinone normally does not contribute significantly to the TOC in mass, and is not the focus of our study. Most likely, other major functional groups, such as carboxylic and carbonyl functional groups, cannot be reduced by dithionite based on their reduction potentials (Bar-Even et al., 2012; Mayhew et al., 1978). Organic sulfur speciation is beyond the scope of this research. Further investigations are needed to elaborate the detailed influences of dithionite reduction on the molecular properties of organic matter. We will add relevant discussion.*

Minor remarks: Line 19. Having this number in the abstract somewhat skews the results of this study towards a higher importance of iron minerals than it really is. Fig. 1 gives a totally different picture. Maybe report median or range here.

*We will change the statement to:*
*We found that on average Fe-bound OC contributed 37.8% of total OC (TOC) in forest soils.*

Lines 90-91: Coarse material was not removed? What about roots?

*Roots and visible plant material were removed. Large particles (>2 mm) were removed by dry sieving. The sentence will be changed accordingly.*

Lines 119-121: Are manganese oxides also affected? Has Mn been measured in addition to Fe?

*Manganese oxides are likely also reduced by dithionite based on their reduction potentials. In soil, generally the Mn content was two orders of magnitude lower than that of Fe (Sparks, 2011). Total electrons, which can be accepted by $MnO_2$, is 10% of that for iron oxides (Lovley, 2011). Although there is rare information about the sorption of OC on Mn, OC bound to Mn would be very likely much less than that associated with Fe oxides. In this study, we did not analyze the influences of manganese oxides on the OC stock.*

Lines 130-131: This is basically the ion exchangeable organic carbon. I probably missed it, but why was this done? Was the exchangeable organic carbon subtracted from the chemically extracted carbon? If not, please discuss why.

*Yes. We subtracted the ion exchangeable OC from the DCB-extractable OC, for determination of the Fe-bound OC.*

Line 151: Why was iron only determined at a single wavelength of 259.9 nm? Using multiple wavelengths has the advantage of identifying interferences that might occur on one wavelength, which can than easily be corrected.

*We picked up the wavelength based on the best standard curve established.*

Lines 160-175 and 309-331: I am not an expert on NEXAFS and FTIR. I suggest the authors and the editor to refer to another reviewer with more expertise on these methods.

*We worked with Canadian beamline scientist to ensure the interpretation and discussion for NEXAFS data.*

Line 180: Are numbers reported as one or two standard deviations of replicates? If so, how many (eg, n = 3)? Are this analytical or sampling replicates?

*All results are based on duplicated sampling sites.*

Lines 187-192: This part is speculative. Since not all soil types are covered the comparison might not hold up at the global level. For example, organic rich soils such as Histosols and Chernozems were not as much considered as they should be based on their global distribution. Further, since the North American distribution of soils differs from the global distribution in that Cambisols, Regosols and iron-rich chromic luvisols are underrepresented, this section should be removed.

*We acknowledge limited types of soils were sampled and analyzed in this study. It is still good information for evaluation of carbon cycles on the large scale. We will shorten the discussion to:*

*Based on an estimate that 1502 Pg (Pg=1×10$^{15}$ g) of OC is stored in terrestrial soils (Scharlemann, et al., 2014), scaling up these results to a global estimate would yield 538.5±271.5 Pg of Fe-bound OC residing in terrestrial soils.*

Lines 202-205: Why was that a surprise? Was it one the main hypothesis of the study? If so, this might be stated in the introduction or methods. After all, shouldn't non-reactive iron store more organic carbon than reactive iron?

*We speculated the fraction of Fe-bound OC would be correlated to the amount of reactive Fe, DCB extractable Fe or free Fe oxides. To minimize confusion, we will revise the statement:*

*There was no significant correlation between $f_{Fe-OC}$ and the concentration of reactive Fe (Pearson Correlation Coefficient r=–0.418, p=0.137, Fig. 2B). This suggests that the proportion of Fe-bound OC is not strongly controlled by the reactive Fe concentration.*

Line 214: How do you define "most of the"? It might be better to state "2/3 of the" or some equally (quantitative) phrase.

*We will change "most of the" to "2/3 of the".*

Lines 241-262: Climate actually controls the type of soil and the type of iron minerals directly through controlling the weathering of iron-bearing primary minerals and the formation of secondary iron-phases. So 6.6°C is probably the ideal condition for soil organic matter production and the formation OM-binding FeOOH minerals. I like to hear a statement by the authors on why "Fe-OC interactions may be particularly pronounced" at 6.6 °C. Can there really be a temperature control on this type of interaction? If so, please provide references for this.

*Very insightful suggestion. We will add discussion:*

*Temperature dependence of Fe-bound OC was regulated by the effects of temperature on the mineral phase of Fe oxides and OC dynamics. Given that ferrihydrite can incorporate more OC than other crystalline Fe oxides, an increase in temperature favors the transformation of ferrihydrite to other crystalline iron oxides (Gnanaprakash et al., 2007; Zhao et al., 1994). However, increase in temperature can also accelerate weathering of other minerals, and increased release of silicon can retard the transformation of ferrihydrite (Cornell et al., 1987; White and Blum, 1995). However, there is also evidence that temperature can affect the chemical composition of soil OC substantially (Conant et al., 2011). For example, increased temperature decreased the content of oxidized functional groups, such as saccharides, which would consequently inhibit the interactions between OC and Fe oxides (Amelung et al., 1997). The overall pattern can be resulted from combined effects of temperature on Fe mineral phase and OC transformation. Further investigations are required to elucidate the mechanism more accurately.*

Lines 252-254: This sentence is somewhat unclear. How does temperature increase the amount of iron bound organic matter (see comment above)? Please re-write.

*The sentence will be re-phrased to:*
*The concentration of Fe-bound OC and $f_{Fe-OC}$ reached peak value at 6.6˚C.*

Line 258: I did expect this as these are relatively "old" soils. The long aging process accumulates old and possibly recalcitrant organic matter along with very stable iron minerals… Also, aging results in more stable iron-organic matter associations as shown by Kaiser et al 2007 (doi:10.2136/sssaj2006.0189).

*We have limited information about aging. But our $^{14}C$ data indicates the age of soils (AL and TK) ranged from 1993 to 2008 (modern soil) (Zhao, unpublished). We did not observe old soils.*

Lines 282-284: …and maybe the reduced chemical potential in small pores of clay-rich soils also limits microbial degradation of soils organic matter (Riedel & Weber 2016, doi:10.2136/sssaj2015.02.0085).

*Good suggestion. We will revise the discussion to:*

*Eusterhues et al. (2005) found a relationship between the resistance of organic matter to oxidative degradation and the clay concentration in soils, suggesting the importance of clay minerals in the stabilization and accumulation of soil OC. Reduced chemical potential of soil organic matter in small pores of clay-rich soils also limits its microbial degradation and enhance its stabilization (Riedel and Weber, 2016). In contrast, Percival et al. (2000) found that the clay mineral fraction explained little of the variation in the accumulation of OC across a range of soil types in New Zealand.*

Lines 304-306: This short sentence is unclear. Please explain this 'concept model' in more detail.

*We will revise the statement and explain the concept model more clearly:*

*This is consistent with Schwertmann et al.1986, who found that the major form of Fe would change from FeOx to complexes with OC when there is higher OC supply.*

Lines 347-349: A nice result. We have observed a similar effect using a complementary method (ultra-high resolution mass spectrometry) in a recent study (doi:10.1016/j.org geochem.2014.02.003). During a leaching test a soil column had turned anoxic and the reductive dissolution of iron- and manganese oxides lead to an increase in the abundance of aromatic compounds leaching from the soil columns. Besides polyphenolic compounds we also observed a release of iron-bound condensed aromatics leading me to belief that iron minerals probably play an important role in sequestering 'black carbon'-type compounds in soils. Do the NEXAFS or FTIR data of this study allow identification of this type of organic matter?

*Very interesting results. We are planning to measure the Fe-bound black carbon in these forest soils, using the quantification of thermo analysis. From NEXAFS and FTIR, it is challenging to unambiguously identify the contribution of black carbon. We will add relevant discussion after line 331:*

*This result was consistent with a previous study using ultra-high resolution mass spectrometry, showing the release of more aromatic carbon during the reductive dissolution of Fe oxides (Riedel et al., 2014).*

Lines 385 and 387. "O/N-rich". This is written like a ratio, but that is not meant here, right?

*We meant rich in both O and N. We will change it to "O&N-rich"*

Lines 382-385. This probably indicates that the lipids are derived from microbial biomass which usually has a lighter isotopic value.

*Good point. We will revise the discussion to:*

*Previous studies shown that $^{13}C$-enriched organic matter in sediments was enriched with O and N (due to the presence of compounds such as proteins and carbohydrate groups), while the microbial biomass-derived lipid fraction was relatively $^{13}C$-depleted (Wang et al., 1998; Zelles et al., 1992).*

Lines 391-392: But it could also be vice versa. Suppose compounds with heavier isotopic composition preferentially sorb to iron.

*Yes. It is difficult to determine the driving force. We will revise the statement to:*

*These relationships suggest that the enrichment in $^{13}C$ was to some degree related to the OC:Fe ratio.*

Lines 392: Where was this "discussed previously"?

*We will change it to:*

*As discussed previously (session 3.2)*

Lines 403-404. A reference would be good here.

*We will add citation of (Yang et al., 2012; Barber et al., 2014).*

*Yang, W. H., Weber, K. A., & Silver, W. L. (2012). Nitrogen loss from soil through anaerobic ammonium oxidation coupled to iron reduction. Nature Geoscience, 5(8), 538-541.*

*Barber, A., Lalonde, K., Mucci, A., & Gélinas, Y. (2014). The role of iron in the diagenesis of organic carbon and nitrogen in sediments: A long-term incubation experiment. Marine Chemistry, 162, 1-9.*

Table 1: I had a hard time understanding table 1, because I am not familiar with the American soil taxonomy. In order to make the paper more readable to a readership beyond North America it might be a good idea if the authors could change to the international FAO soil classification standard, if this is possible. After all the paper has been submitted to a journal run by the 'European' Geophysical Union…

*We will add FAO classification.*

Figure 3: What was the hypothesis behind the linear regression analysis of TOC vs latitude? Did the authors suspect a linear relationship? If so, please explain why.

*We have relevant discussion in the text (line 235-237):*
*There was a significant correlation between the TOC concentration and latitude (Pearson correlation coefficient p=0.619, r=0.018), a pattern commonly observed due to lower microbial activity and turnover rates of C at higher, colder latitudes (Davidson and Janssens, 2006).*
*We only discussed the correlation, rather than regression. To avoid confusion, we will remove the line.*

**Table 1 Information for the 14 forest sites studied (Obrist et al., 2011, 2012, 2015)**

| Forest ID | Abbr. | Location | Soil Order (US) | Soil Class[a] (FAO) | Climate Zone | Precip.[b] (mm y$^{-1}$) | Temp[c] (°C) | LAT(°)[d] | LONG(°)[e] | Elevation (m asl) |
|---|---|---|---|---|---|---|---|---|---|---|
| Ashland | **AL** | Ashland, Missouri | Alfisols | Luvisols & Greyzems | Humid Continental | 1023 | 13.9 | 38.73 | -92.20 | 210 |
| Bartlett | **BL** | Bartlett, New Hampshire | Spodosols | Podzols & Lithosols | Humid Continental | 1300 | 4.5 | 44.0 | -71.29 | 272 |
| Marysville | **MS** | Marysville, California | Mollisols | Luvisols | Mediterranean climate | 775 | 16.9 | 39.25 | -121.28 | 386 |
| Gainesville | **GS** | Gainesville, Florida | Spodosols | Podzols | Humid Subtropical | 1228 | 21.7 | 29.74 | -82.22 | 50 |
| Oak Ridge | **OR** | Oak Ridge, Tennessee | Ultisols | Acrisols | Humid Subtropical | 1350 | 14.5 | 35.97 | -84,28 | |
| Little Valley (post-fire) | **LVF** | Little Valley, Nevada | Entisols | Arenosols | Highland Climate | 551 | 5.0 | 39.12 | -119.93 | 2010 |
| Little Valley | **LV** | Little Valley, Nevada | Entisols | Arenosols | Highland Climate | 550 | 5.0 | 39.12 | -119.93 | 2011 |
| Truckee (post-fire) | **TKF** | Truckee, California | Alfisols | Luvisols | Highland Climate | 569 | 6.0 | 39.37 | -120.1 | 1768 |
| Truckee | **TK** | Truckee, California | Alfisols | Luvisols | Highland Climate | 568 | 5.9 | 39.37 | -120.1 | 1767 |
| Niwot Ridge | **NR** | Niwot Ridge, Colorado | Alfisols | Cambisols | Highland Climate | 800 | 1.3 | 40.03 | -105.55 | 3050 |
| Hart | **HT** | Hart, Michigan | Spodosols | Podzols | Humid Continental | 812 | 7.6 | 43.67 | -86.15 | 210 |
| Howland | **HL** | Howland, Maine | Spodosols | Luvisols | Humid Continental | 1040 | 6.7 | 45.20 | -68.74 | 60 |
| Thompson I | **TSI** | Ravensdale, Washington | Inceptisols | Cambisols | Highland Climate | 1141 | 9.8 | 47.38 | -121.93 | 221 |
| Thompson II | **TSII** | Ravensdale, Washington | Inceptisols | Cambisols | Highland Climate | 1140 | 9.8 | 47.38 | -121.93 | 220 |

a: Food and Agriculture Organization; b: annual precipitation; c: annual mean temperature; d latitude; e: longitude.

References:

Adhikari, D., Poulson, S. R., Sumaila, S., Dynes, J. J., McBeth, J. M., and Yang, Y.: Asynchronous reductive release of iron and organic carbon from hematite–humic acid complexes, Chemical Geology, 430, 13-20, 2016.

Amelung, W., Flach, K. W., and Zech, W.: Climatic effects on soil organic matter composition in the great plains, Soil Science Society of America Journal, 61(1), 115-123, 1997.

Barber, A., Lalonde, K., Mucci, A., and Gélinas, Y.: The role of iron in the diagenesis of organic carbon and nitrogen in sediments: A long-term incubation experiment, Marine Chemistry, 162, 1-9, 2014.

Bar-Even, A., Flamholz, A., Noor, E., and Milo, R.: Rethinking glycolysis: on the biochemical logic of metabolic pathways, Nature Chemical Biology, 8(6), 509-517, 2012.

Borggaard, O. K.: Influence of iron oxides on the surface area of soil. J. Soil Sci., 33, 443–449, 1982.

Conant, R. T., Ryan, M. G., Agren, G. I., Birge, H. E., Davidson, E. A., Eliasson, P. E., Evans, S. E., Frey, S. D., Giardina, C. P., Hopkins, F. M., Hyvonen, R., Kirschbaum, M. U. F., Lavallee, J. M., Leifeld, J., Parton, W. J., Steinweg, J. M., Wallenstein, M. D., Wetterstedt, J. A. M., and Bradford, M. A.: Temperature and soil organic matter decomposition rates - synthesis of current knowledge and a way forward, Global Change Biology, 17(11), 3392-3404, 2011.

Coffin, D. E.: A method for the determination of free iron in soils and clays, Canadian Journal of Soil Science, 43, 7–17, 1963.

Cornell, R. M., Giovanoli, R., and Schindler, P. W.: Effect of silicate species on the transformation of ferrihydrite into goethite and hematite in alkaline media, Clays and Clay Minerals, 35(1), 21-28, 1987.

Davidson, E. A., and Janssens, I. A.: Temperature sensitivity of soil carbon decomposition and feedbacks to climate change, Nature, 440(7081), 165-173, 2006.

Gnanaprakash, G., Mahadevan, S., Jayakumar, T., Kalyanasundaram, P., Philip, J., and Raj, B.: Effect of initial pH and temperature of iron salt solutions on formation of magnetite nanoparticles, Materials Chemistry and Physics, 103(1), 168-175, 2007.

Harden, J. W., Sundquist, E. T., Stallard, R. F., and Mark, R. K.: Dynamics of soil carbon during the deglaciation of the Laurentide ice sheet, Science, 258, 1921–1924, 1992.

Lalonde, K., Mucci, A., Ouellet, A., and Gélinas, Y.: Preservation of organic matter in sediments promoted by iron, Nature, 483(7388), 198-200, 2012.

Lovley, D. R.: Live wires: direct extracellular electron exchange for bioenergy and the bioremediation of energy-related contamination. Energy & Environmental Science, 4(12), 4896-4906, 2011.

Mayhew, S. G.: The redox potential of dithionite and $SO^{-2}$ from equilibrium reactions with flavodoxins, methyl viologen and hydrogen plus hydrogenase, European Journal of Biochemistry, 85(2), 535-547, 1978.

Page, S. E., Siegert, F., Rieley, J. O., Boehm, H. D. V., Jaya, A., and Limin, S.: The amount of carbon released from peat and forest fires in Indonesia during 1997, Nature, 420(6911), 61-65, 2002.

Pinheiro-Dick, D., and Schwertmann, U.: Microaggregates from Oxisols and Inceptisols: dispersion through selective dissolution and physicochemical treatments, Geoderma, 74, 49–63, 1996.

Riedel, T., Iden, S., Geilich, J., Wiedner, K., Durner, W., and Biester, H.: Changes in the molecular composition of organic matter leached from an agricultural topsoil following addition of biomass-derived black carbon (biochar), Organic Geochemistry, 69, 52-60, 2014.

Riedel, T., and Weber, T. K.: The chemical potential of water in soils and sediments, Soil Science Society of America Journal, 80(1), 79-83, 2016.

Scharlemann, J. P. W., Tanner, E. V. J., Hiederer, R., and Kapos, V.: Global soil carbon: understanding and managing the largest terrestrial carbon pool, Carbon Management, 5, 81-91, 2014.

Schwertmann, U., and Murad, E.: The nature of an iron oxide–organic iron association in a peaty environment. Clay Miner., 23, 291–299, 1988.

Sparks, D. L.: Environmental Soil Chemistry, Second Edn., edited by: Sparks, D. L., Academic Press, Cambridge, MA, 2002.

Wagai, R., and Mayer, L. M.: Sorptive stabilization of organic matter in soils by hydrous iron oxides, Geochimica et Cosmochimica Acta, 71(1), 25-35, 2007.

Wang, X. C., Druffel, E. R. M., Griffin, S., Lee, C., and Kashgarian, M.: Radiocarbon studies of organic compound classes in plankton and sediment of the northeastern Pacific Ocean, Geochimica Et Cosmochimica Acta, 62, 1365-1378, 1998.

White, A. F., and Blum, A. E.: Effects of climate on chemical-weathering in watersheds, Geochimica Et Cosmochimica Acta, 59(9), 1729-1747, 1995.

Yang, W. H., Weber, K. A., and Silver, W. L.: Nitrogen loss from soil through anaerobic ammonium oxidation coupled to iron reduction. Nature Geoscience, 5(8), 538-541, 2012.

Zelles, L., Bai, Q. Y., Beck, T., and Beese, F.: Signature fatty acids in phospholipids and lipopolysaccharides as indicators of microbial biomass and community structure in agricultural soils, Soil Biology and Biochemistry, 24(4), 317-323, 1992.

Zhao, J. M., Huggins, F. E., Feng, Z., and Huffman, G. P.: Ferrihydrite-surface-structure and its effects on phase-transformation, Clays and Clay Minerals, 42 (6), 737-746, 1994.

---

## Author Response (AR2)

August 3, 2016

**Response to Comments**

**Comments are in black font and our responses are in *italicized* font.**

**Associate Editor**

Dear Authors,

Even without looking in detail at the main manuscript, more work is needed. I would start with the Table and Figures. Normally these are self-explanatory or mostly, some of yours lack this. These are my suggestions for improvement.

*Thank you very much for your helpful suggestions. We have revised the manuscript accordingly.*

Table 1 I would not put the 14 sites in alphabetically order, but select a parameter relevant to the explaining the observations in your work, e.g. you could list them in order of low to high precipitation or temperature or the other way.

*Based on the correlation in Fig. 3, we have re-ordered them according to latitude. Please see Table 1 for revision.*

Fig 1. The placing of the locations and displayed parameters on the USA map is nice, but why do I need to look at them on a map? Is there more C up North more bound Fe-C, maybe indicate what it shows with a view arrows .... is it related to T or precipitation, see comment Table 1 maybe this is the parameter to list Table 1 under.

*As similar as for Table 1, there is generally higher TC and TOC in soils with higher latitude. We feel it is good to keep them in the map. We revised the figure caption accordingly to emphasize the trend.*

Fig 2 what relationship are shown, if the are there they are not obvious. If there are no relationship, why put in a figure at all.

*Fig. 2A shows the contents of reactive iron and OC:Fe molar ratio. Fig. 2B shows the variations in OC:Fe, discussed in the manuscript. Fig. 2B was moved to the supplementary material.*

Fig 3 Do we need all the show subfigures? I would prefer a selection of the most pertinent examples which link to the text, with the rest going to supplementary material.

*Partial of Fig. 3 was moved to supplementary material.*

Fig 4 This is very poor Figure, messy and unclear it gives a bad impression of overall standard of the manuscript, it lets your paper down.

*This figure has been improved accordingly. The relationships between labile carbon/uncalibrated Fe-bound organic carbon and texture were kept. Relationship between Fe-bound organic carbon and texture was moved to supplementary material, as there is no significant correlation.*

Fig 5 see comment Fig 3

*We kept the spectra for aliphatic carbon in the manuscript, and moved other spectra to supplementary material.*

Fig 6. A lot of information but no self-evident message from both Figures, please condense the information more and better.

*This figure has been improved by keeping the $^{13}C$ for Fe-bound OC and non-Fe-bound OC in the manuscript, and moving other data to the supplementary material. Now, it clearly showed the enrichment of $^{13}C$ in Fe-bound OC.*

[revised manuscript text omitted]